# Both consumptive and non-consumptive effects of predators impact mosquito populations and have implications for disease transmission

**Marie C Russell[1]\*[†], Catherine M Herzog[2†], Zachary Gajewski[3], Chloe Ramsay[4], Fadoua El Moustaid[3], Michelle V Evans[5,6], Trishna Desai[7], Nicole L Gottdenker[5,8], Sara L Hermann[9], Alison G Power[10], Andrew C McCall[11]**

[1]Department of Life Sciences, Imperial College London, Silwood Park Campus, Ascot, United Kingdom; [2]Center for Infectious Disease Dynamics, Pennsylvania State University, University Park, United States; [3]Department of Biological Sciences, Virginia Polytechnic Institute and State University, Blacksburg, United States; [4]Department of Biological Sciences, University of Notre Dame, Notre Dame, United States; [5]Odum School of Ecology & Center for Ecology of Infectious Diseases, University of Georgia, Athens, United States; [6]MIVEGEC, IRD, CNRS, Université Montpellier, Montpellier, France; [7]Nuffield Department of Population Health, University of Oxford, Oxford, United Kingdom; [8]Department of Veterinary Pathology, University of Georgia College of Veterinary Medicine, Athens, United States; [9]Department of Entomology, Pennsylvania State University, University Park, United States; [10]Department of Ecology & Evolutionary Biology, Cornell University, Ithaca, United States; [11]Biology Department, Denison University, Granville, United States

**\*For correspondence:**
marie.clare.russell@gmail.com

[†]These authors contributed equally to this work

**Competing interest:** The authors declare that no competing interests exist.

**Abstract** Predator-prey interactions influence prey traits through both consumptive and non-consumptive effects, and variation in these traits can shape vector-borne disease dynamics. Meta-analysis methods were employed to generate predation effect sizes by different categories of predators and mosquito prey. This analysis showed that multiple families of aquatic predators are effective in consumptively reducing mosquito survival, and that the survival of *Aedes*, *Anopheles*, and *Culex* mosquitoes is negatively impacted by consumptive effects of predators. Mosquito larval size was found to play a more important role in explaining the heterogeneity of consumptive effects from predators than mosquito genus. Mosquito survival and body size were reduced by non-consumptive effects of predators, but development time was not significantly impacted. In addition, *Culex* vectors demonstrated predator avoidance behavior during oviposition. The results of this meta-analysis suggest that predators limit disease transmission by reducing both vector survival and vector size, and that associations between drought and human West Nile virus cases could be driven by the vector behavior of predator avoidance during oviposition. These findings are likely to be useful to infectious disease modelers who rely on vector traits as predictors of transmission.

## Editor's evaluation

This careful meta-analysis evaluates consumptive and non-consumptive effects of aquatic predators across multiple mosquito species, drawing from laboratory and semi-field studies. The authors find an important role for larval size in moderating consumption, significant non-consumptive impacts of predators on survival and body size, and variable effects of predators on oviposition behavior. These

results therefore highlight multiple mechanisms by which aquatic predators might affect disease transmission.

## Introduction

While it is well known that predation reduces vector populations through consumptive effects, non-consumptive effects of predators can also greatly impact prey demographics (*Preisser et al., 2005*). Mosquitoes are vectors of a variety of debilitating and deadly diseases, including malaria, lymphatic filariasis, and arboviruses, such as chikungunya, Zika, and dengue (*Weaver and Reisen, 2010*; *WORLD HEALTH ORGANIZATION, 2020*). Consequently, there is motivation from a public health perspective to better understand the different drivers of variation in mosquito traits that can ultimately impact vector population growth and disease transmission. In addition, recent work has suggested that incorporation of vector trait variation into disease models can improve the reliability of their predictions (*Cator et al., 2020*). In this study, systematic review and meta-analysis methods are used to synthesize a clearer understanding of the consumptive and non-consumptive effects of predators on mosquito traits, including survival, oviposition, development, and size.

Mosquito insecticide resistance is recognized as a growing problem (*Hancock et al., 2018*; *Hemingway and Ranson, 2000*; *Liu, 2015*) leading some to suggest that control efforts should rely more heavily on 'non-insecticide based strategies' (*Benelli et al., 2016*). The consumptive effects of predators on mosquitoes have previously been harnessed for biocontrol purposes. Past biocontrol efforts have used predators such as cyclopoid copepods (*Kay et al., 2002*; *Marten, 1990*; *Russell et al., 1996*; *Veronesi et al., 2015*) and mosquitofish (*Pyke, 2008*, *Seale, 1917*) to target the mosquito's aquatic larval stage. The strength of the consumptive effects of these predators on mosquitoes can be influenced by multiple factors, including predator-prey size ratio and temperature. Predator-prey body size ratios tend to be higher in freshwater habitats than other types of habitats (*Brose et al., 2006*), and attack rate tends to increase with temperature (*Kalinoski and DeLong, 2016*; *Dam and Peterson, 1988*), although other studies suggest a unimodal response to temperature (*Uiterwaal and Delong, 2020*; *Englund et al., 2011*).

Predators can also have non-consumptive effects on prey (*Peacor and Werner, 2001*), and these effects are thought to be more pronounced in aquatic ecosystems than in terrestrial ecosystems (*Preisser et al., 2005*). Non-consumptive effects of predators are the result of the prey initiating anti-predator behavioral and/or physiological trait changes that can aid in predator avoidance (*Hermann and Landis, 2017*; *Lima and Dill, 1990*). Such plasticity in certain prey traits may also result in energetic costs (*Lima, 1998*). Predator detection is key for these trait changes to occur and can be mediated by chemical, tactile, and visual cues (*Hermann and Thaler, 2014*). In mosquitoes, exposure to predators is known to affect a variety of traits including behavior, size, development, and survival (*Arav and Blaustein, 2006*; *Bond et al., 2005*; *Roberts, 2012*; *Roux et al., 2015*, *Zuharah et al., 2013*). Experimental observations of predator effects on mosquito size and development are inconsistent and results sometimes vary by mosquito sex. For example, exposure to predation was found to increase the size of *Culex pipiens* mosquitoes (*Alcalay et al., 2018*) but decrease the size of *Culiseta longiareolata* (*Stav et al., 2005*). In addition, female *Aedes triseriatus* exhibited shorter development times when exposed to predation at high nutrient availability (*Ower and Juliano, 2019*), but male *C. longiareolata* had longer development times in the presence of predators (*Stav et al., 2005*). In some cases, a shared evolutionary history between predator and prey organisms can strengthen the non-consumptive effects of predators on mosquitoes (*Buchanan et al., 2017*; *Sih, 1986*).

This investigation assesses the consumptive and non-consumptive effects of predators on mosquito traits and describes how these effects could impact disease transmission. The roles of vector genus, predator family, mosquito larval instar (an indicator of prey size), and temperature are also examined as potential moderators of predator effects. Non-consumptive effects of predators are expected to cause a smaller reduction in mosquito survival than consumptive effects because, in practice, measures of consumptive effects always include both consumptive and non-consumptive effects. Based on previous findings, larger predators are more likely to consumptively reduce mosquito survival (*Kumar et al., 2008*). In addition, *Aedes* mosquito larvae may be more vulnerable to consumption than other genera because of the high degree of motility observed in this genus (*Dieng et al., 2003*; *Marten and Reid, 2007*; *Soumare and Cilek, 2011*). The oviposition response to predation is expected to

**eLife digest** Mosquitoes are often referred to as the deadliest animals on earth because some species spread malaria, West Nile virus or other dangerous diseases when they bite humans and other animals. Adult mosquitoes fly to streams, ponds and other freshwater environments to lay their eggs. When the eggs hatch, the young mosquitoes live in the water until they are ready to grow wings and transform into adults.

In the water, the young mosquitoes are particularly vulnerable to being eaten by dragonfly larvae, fish and other predators. When adult females are choosing where to lay their eggs, they can use their sense of smell to detect these predators and attempt to avoid them. Along with eating the mosquitoes, the predators may also reduce mosquito populations in other ways. For example, predators can disrupt feeding among young mosquitoes, which may affect the time that it takes for them to grow into adults or the size of their bodies once they reach the adult stage. Although the impacts of different predators have been tested separately in multiple settings, the overall effects of predators on the ability of mosquitoes to spread diseases to humans remain unclear.

To address this question, Russell, Herzog et al. used an approach called meta-analysis on data from previous studies. The analysis found that along with increasing the death rates of mosquitoes, the presence of predators also leads to a reduction in the body size of those mosquitoes that survive, causing them to have shorter lifespans and fewer offspring.

Russell, Herzog et al. found that one type of mosquito known as *Culex* – which carries West Nile virus – avoided laying its eggs near predators. During droughts, increased predation in streams, ponds and other aquatic environments may lead adult female *Culex* mosquitoes to lay their eggs closer to residential areas with fewer predators. Russell, Herzog et al. propose that this may be one reason why outbreaks of West Nile virus in humans are more likely to occur during droughts.

In the future, these findings may help researchers to predict outbreaks of West Nile virus, malaria and other diseases carried by mosquitoes more accurately. Furthermore, the work of Russell, Herzog et al. provides examples of mosquito predators that could be used as biocontrol agents to decrease numbers of mosquitoes in certain regions.

be weakest among *Aedes* species that oviposit above the water line, due in part to their delayed-hatching eggs (*Vonesh and Blaustein, 2010*). Predation is predicted to reduce mosquito size and lengthen development time, consistent with the reduced growth response observed in other insect systems (*Hermann and Landis, 2017*). Certain non-consumptive effects of predation, particularly oviposition site selection and decreased vector size, are likely to play important roles in the dynamics of mosquito-borne disease.

## Materials and methods
### Literature screening
A systematic search was conducted for studies on predation of mosquitoes that were published between 1970 and July 1, 2019 using both PubMed and Web of Science search engines, according to the PRISMA protocol (*Moher et al., 2009*). Mosquito vectors of the *Anopheles* and *Aedes* genera were specifically highlighted in our search terms because these genera contain the vector species that transmit malaria, yellow fever, and dengue – the three most deadly mosquito-borne diseases worldwide (*Hill et al., 2005*). Searches included 18 combinations of three vector predation terms (mosquito predat*, *Anopheles* predat*, *Aedes* predat*) and six trait terms (survival, mortality, development, fecundity, dispers*, host preference). Abstracts from the 1136 studies were each screened by two different co-authors, using the 'metagear' package in R (*Lajeunesse, 2016*, *R Development Core Team, 2020*). If either screener thought the study had information relevant to predation of mosquitoes, or both screeners thought the abstract was ambiguous, the study was read in full. This resulted in 306 studies that were fully reviewed to determine if any predation data could be extracted (*Figure 1*).

### Study exclusion criteria
Data were extracted from studies that collected data on non-consumptive and/or consumptive effects of predators on mosquitoes. Studies were required to have a mean, error measurement, and at least

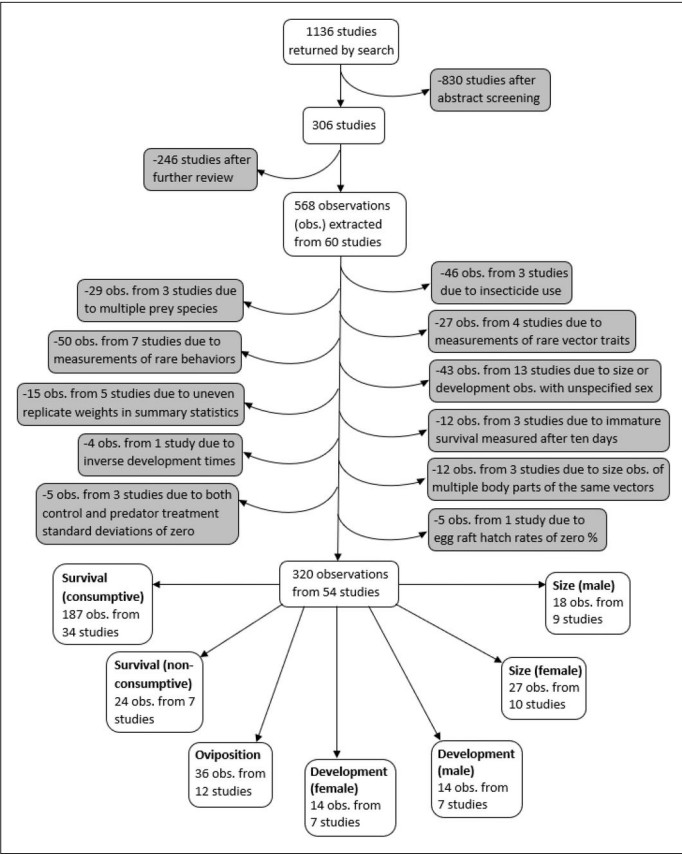

**Figure 1.** Flowchart demonstrating the literature search, screening process, data exclusions, and the resulting seven different vector trait data subsets.

two replicates for both control and predator treatments. The control treatment was required to have all the same conditions as the predator treatment, such as prey density and type of water, without the predators. Studies that were not published in English and studies that did not differentiate between predators of multiple families were excluded. Studies were also excluded if oviposition by free-flying female mosquitoes could have interfered with observing the consumptive effects of predators on vector survival. The final database comprised data extracted from 60 studies (*Supplementary file 1*). The data included observations from laboratory experiments, as well as semi-field experiments, in which mesocosms of different treatments were observed in outdoor settings.

## Data extraction

Variables related to the publication, the vector, the predator, and the effect size (*Table 1*) were extracted from each study. Data from tables and text were recorded as they were published, and data from figures were extracted using WebPlotDigitizer (*Rohatgi, 2020*). Error measurements that were not originally presented as standard deviations were converted to standard deviations prior to the effect size calculation.

## Data exclusions

A PRISMA plot of literature inclusion and exclusion is provided in *Figure 1*. Observations where insecticide was used were excluded because insecticides are known to interfere with consumptive and non-consumptive effects of predators (*Delnat et al., 2019*; *Janssens and Stoks, 2012*). In addition, observations from experiments with mosquito prey of two or more species were excluded because it was not possible to account for effects from apparent competition or prey-switching. Observations of vector fecundity, vector competence, behavioral traits other than oviposition, as well as observations where the vector trait was marked as 'other' were not analyzed because each of these traits were only recorded from three or fewer studies.

**Table 1.** Variables extracted from included studies.

| Variable | Description |
| --- | --- |
| Publication data: | |
| Title | Full study title |
| Journal | Name of journal that published the study |
| Year | Year of publication |
| Study environment | Environment where the experiment took place: lab or semi-field |
| Vector data: | |
| Order, Family, Genus, Species | Taxonomic identification |
| Trait | Outcome that was measured (e.g. survival, development, etc.) |
| Stage | Life stage: egg, larva, pupa, or adult |
| Larval instar | Early (1st and 2nd instars), late (3rd and 4th instars), both, or NA (eggs, pupae, or adults) |
| Sex | Male or female |
| Predator data: | |
| Phylum, Class, Order, Family, Genus, Species | Taxonomic identification |
| Starved | Whether the predator was starved: yes or no |
| Time starved | Amount of time that the predator was starved (in minutes) |
| Predation effect | Consumptive or non-consumptive |
| Effect size data: | |
| Units | Units of extracted data |
| Control mean | Average of the outcome measured among the controls |
| Control standard deviation | Standard deviation of the outcome measured in the controls |
| Control number of replicates | Number of control replicates |
| Predation mean | Average of the outcome measured in the predator treatment |
| Predation standard deviation | Standard deviation of the outcome measured in the predator treatment |
| Predation number of replicates | Number of predation replicates |
| Experiment ID | Alphabetic assignment to mark observations sharing a control group or representing the same prey individuals as originating from the same experiment |
| Additional data: | |
| Experiment time (days) | Duration of the experiment in days |
| Data source | Graph or text |
| Number of predators | Number of predators with access to prey, or 'cue' if there are no predators with direct access to prey |
| Number of prey (vectors) | Number of mosquito prey that are exposed to predation |

*Table 1 continued on next page*

*Table 1 continued*

| Variable | Description |
| --- | --- |
| Arena volume (mL) | Volume of the arena where prey encounter predators |
| Time exposed to predator(s) | Amount of time (in days) when the predator has direct access to the mosquito prey |
| Temperature (°C) | Temperature during the predation interaction |
| Type of predator cue | Predator cues, or cues from both predator(s) and dying conspecifics; NA for observations with a consumptive predation effect |

Due to protandry, the earlier emergence of males to maximize their reproductive success, mosquitoes respond to sex-specific selective forces that influence their development time and body size (*Kleckner et al., 1995*). Under low resource conditions, female mosquitoes are likely to maximize body mass by extending their development time, whereas males tend to minimize their development time at the expense of lower body mass (*Kleckner et al., 1995*). Observations of mosquito development time and body size in our database that were not sex-specific were excluded so that these vector traits could be analyzed while controlling for sex. In addition, among the observations of development time and body size, some predator means did not necessarily represent an evenly weighted average of the replicates. For example, if a total of 20 mosquitoes from three different predator replicates survived to adulthood, the mean development time and size of those 20 individuals may have been reported. To represent an evenly weighted average of the replicates, it is necessary to first calculate summary statistics among multiple individuals that emerge from the same replicate, and then report the average of the replicate-specific means. Observations that might have been influenced by uneven representation of replicates were excluded to prevent pseudo-replication from altering later meta-analyses.

For consumptive observations where life stage-specific survival was reported after more than 10 days of predator exposure, only data on survival marked by adult emergence were included for analysis. Effects observed among immature vector stages after such a long period of predator exposure were not analyzed because they could have resulted from a combination of non-consumptive effects on development, and consumptive effects on survival. Development time observations that were reported as the inverse of development time (units of days$^{-1}$) were excluded because although their means could be converted to units of days, their standard deviations could not be converted to match units of days. In cases where multiple body sections of the same mosquitoes were measured to produce multiple size observations, only the wing measurement was included in the analysis to prevent pseudo-replication. Observations in which both the control and the predator treatments had standard deviations of zero were excluded because the meta-analysis methods did not support non-positive sampling variances.

## Exclusions and data substitutions for predator treatment means of zero

One study that was included in our database reported egg survival data as the hatch rate of field collected *Culex pervigilans* rafts (*Zuharah et al., 2013*). However, mosquitoes have been shown to lay eggs independent of mating (*O'Meara, 1979*), and hatch rates of zero have previously been observed in rafts laid by *Culex* females that were held separately from males (*Su and Mulla, 1997*). Thus, hatch rates of zero were excluded from further analysis because these values may represent unfertilized egg rafts, rather than a strong impact of predators on survival. Twenty of the 187 consumptive survival observations had a predation mean of zero, and each of these zeros resulted from experiments that began with a specified number of live larvae. Consumptive survival zeros were each replaced with 0.5% of the starting number of mosquito prey to avoid undefined effect sizes. In addition, there was one zero out of the 36 oviposition predation means; this value had units of 'number of egg rafts laid' and was replaced with 0.5 rafts. Similar methods for replacing zero values in the treatment mean with small non-zero values have previously been employed (*Thapa et al., 2018*).

The final analysis dataset included seven subsets: consumptive effects on survival, non-consumptive effects on survival, oviposition, development (female and male), and size (female and male). The

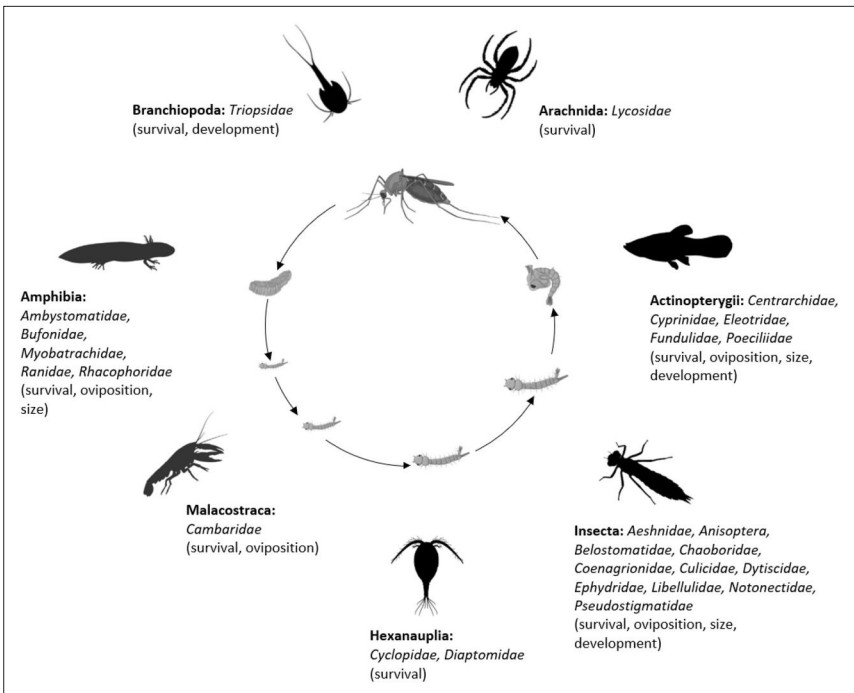

**Figure 2.** Mosquito predator classes (bold font) and families (italicized font) included in the database and the vector traits that they may influence (in parentheses); predator images not to scale, and placed randomly with respect to the different mosquito life stages. Image sources: phylopic.org (CC BY 3.0 or public domain): Actinopterygii (creator: Milton Tan), Arachnida (creators: Sidney Frederic Harmer & Arthur Everett Shipley, vectorized by Maxime Dahirel), Branchiopoda (creator: Africa Gomez), and Insecta (creator: Marie Russell). BioRender.com: Amphibia, Hexanauplia, and Malacostraca class silhouettes; mosquito larval instars, pupa, and blood-feeding adult. Trishna Desai: mosquito egg raft.

data included 187 observations from 34 studies of consumptive survival, 24 observations from seven studies of non-consumptive survival, 36 observations from 12 studies of oviposition, 14 observations from seven studies of female development, 14 observations from seven studies of male development, 27 observations from 10 studies of female size, and 18 observations from nine studies of male size (*Figure 1*). These observations covered seven different classes of predator families (*Figure 2*).

## Data analysis
### Measuring effect sizes and heterogeneity
All analyses were conducted in R version 4.0.2 (*R Development Core Team, 2020*). For each subset of trait data (*Figure 1*), the ratio of means (ROM) measure of effect size was calculated using the 'escalc' function from the 'metafor' package; this effect measure is equal to a log-transformed fraction, where predation mean is the numerator and control mean is the denominator (*Viechtbauer, 2010*). Random effects models, using the 'rma.uni' function, were run with the ROM effect sizes as response variables; each model had a normal error distribution and a restricted maximum likelihood (REML) estimator for $\tau^2$, the variance of the distribution of true effect sizes (*Viechtbauer, 2010*). Although these random effects models could not account for multiple random effects or moderators, they provided overall estimates of the ROM effect sizes and estimates of the $I^2$ statistics. Each $I^2$ statistic represented the percentage of total variation across studies due to heterogeneity (*Higgins et al., 2003*). If the $I^2$ statistic was equal to or greater than 75%, the heterogeneity was considered to be high (*Higgins et al., 2003*), and high heterogeneity has previously motivated further testing of moderators (*Vincze et al., 2017*).

### Assessing publication bias
Publication bias was assessed by visually inspecting funnel plots and conducting Egger's regression test ('regtest' function) with standard error as the predictor (*Sterne and Egger, 2001*; *Viechtbauer,*

*2010*). If the Egger's regression test showed significant evidence of publication bias based on funnel plot asymmetry, the 'trim and fill' method ('trimfill' function) was used to estimate how the predation effect size might change after imputing values from missing studies (*Duval and Tweedie, 2000a*, *Duval and Tweedie, 2000b*; *Viechtbauer, 2010*). The trim and fill method has previously been recommended for testing the robustness of conclusions related to topics in ecology and evolution (*Jennions and Møller, 2002*). Of the two trim and fill estimators, $R_0$ and $L_0$, that were originally recommended (*Duval and Tweedie, 2000a*, *Duval and Tweedie, 2000b*), the $L_0$ estimator was used in this study because it is more appropriate for smaller datasets (*Shi and Lin, 2019*).

## Testing moderators

Data subsets that had high heterogeneity, observations from at least 10 studies, and no evidence of publication bias according to Egger's regression results were analyzed further using multilevel mixed effects models with the 'rma.mv' function (*Viechtbauer, 2010*; *Higgins et al., 2020*). All multilevel mixed effects models had normal error distributions, REML estimators for $\tau^2$, and accounted for two random factors: effect size ID, and experiment ID nested within study ID. Moderators, such as predator family, vector genus, larval instar (directly correlated to prey size), and temperature, were tested within each data subset to determine if they affected the observed heterogeneity in ROM effect sizes. For categorical moderators, the intercept of the multilevel mixed effects model was removed, allowing an analysis of variance (ANOVA) referred to as the 'test of moderators' to indicate if any of the categories had an effect size different than zero. For data subsets with observations from 10 to 29 studies, only one moderator was tested at a time to account for sample size constraints. For subsets with observations from a higher number of studies (30 or more), up to two moderators were tested at once, and interaction between moderators was also tested. The small sample corrected Akaike Information Criterion (AICc) was used to compare multilevel mixed effects models and to select the model of best fit within each data subset; differences in AICc greater than two were considered meaningful (*Burnham and Anderson, 2004*).

# Results
## Random effects models

Each data subset (*Figure 1*) had an $I^2$ statistic of greater than 75%, indicating high heterogeneity (*Higgins et al., 2003*). Random effects model results showed that predators consumptively decreased mosquito survival with an effect size of –1.23 (95% CI –1.43,–1.03), p-value < 0.0001, and non-consumptively reduced survival with a smaller effect size of –0.11 (95% CI –0.17,–0.04), p-value = 0.0016. In addition, predators non-consumptively reduced oviposition behavior with an effect size of –0.87 (95% CI –1.31,–0.42), p-value = 0.0001, and mosquito body size was non-consumptively reduced by predators in both males and females; the female effect size was –0.13 (95% CI –0.19,–0.06), p-value = 0.0002, and the male effect size was –0.03 (95% CI –0.06,–0.01), p-value = 0.0184. There was not a significant non-consumptive effect of predators on either male or female development time; the female effect size was –0.01 (95% CI –0.09, 0.07), p-value = 0.7901, and the male effect size was –0.04 (95% CI –0.12, 0.04), p-value = 0.3273.

The Egger's regression test results showed that the non-consumptive survival subset, both development time subsets (male and female), and the female size subset exhibited funnel plot asymmetry indicative of publication bias. The 'trim and fill' procedure identified missing studies in the non-consumptive survival subset and the female size subset, but the procedure did not identify any missing studies in either of the development time subsets. Three studies were estimated to be missing from the non-consumptive survival data, and accounting for imputed values from missing studies resulted in a shift in the predation effect size from –0.11 (95% CI –0.17,–0.04), p-value = 0.0016, to -0.13 (95% CI –0.20,–0.07), p-value < 0.0001. Two studies were estimated to be missing from the female size data, and accounting for imputed values from these missing studies shifted the predation effect size from –0.13 (95% CI –0.19,–0.06), p-value = 0.0002, to -0.10 (95% CI –0.17,–0.03), p-value = 0.0083. Shifts in effect size estimates due to the trim and fill procedure were minor and did not cause any of the observed effects of predators to change direction or become insignificant.

## Multilevel mixed effects models

The consumptive survival and oviposition data subsets met the criteria of high heterogeneity, observations from at least 10 studies, and no evidence of publication bias. Therefore, these data subsets were tested for moderators using multilevel mixed effects models. Predator families that decreased mosquito survival included Cyprinidae: –3.44 (95% CI –5.79,–1.09), p-value = 0.0042; Poeciliidae: –1.42 (95% CI –2.67,–0.16), p-value = 0.0270; Ambystomatidae: –5.18 (95% CI –7.94,–2.42), p-value = 0.0002; Aeshnidae: –2.93 (95% CI –4.80,–1.07), p-value = 0.0020; and Notonectidae: –2.14 (95% CI –3.07,–1.21), p-value < 0.0001 (*Figure 3a*). Vector genera that experienced significant decreases in survival due to consumptive effects of predators included *Aedes*: –1.23 (95% CI –1.81,–0.65), p-value < 0.0001; *Anopheles*: –1.34 (95% CI –2.01,–0.66), p-value = 0.0001; and *Culex*: –1.41 (95% CI –1.96,–0.86), p-value < 0.0001 (*Figure 3b*). Among all 187 consumptive survival observations from 34 studies, the best model fit, according to AICc value, was achieved when an interaction between predator family and vector genus was included in the model (*Table 2*). However, among the 163 larval stage consumptive survival observations from 30 studies, adding an interactive term between larval instar (an indicator of prey size) and predator family had a greater improvement on model fit than adding an interactive term between vector genus and predator family (*Figure 3c*, *Table 3*). Temperature did not affect the heterogeneity of consumptive survival data, either as a linear moderator: –0.01 (95% CI –0.10, 0.07), p-value = 0.7559, or a quadratic moderator: 0.00 (95% CI 0.00, 0.00), p-value = 0.8184. The best oviposition model fit, according to AICc value, was achieved when vector genus was added as a moderator (*Table 4*). The mean oviposition effect size was not significantly different than zero for *Aedes*: 0.32 (95% CI –2.14, 2.79), p-value = 0.7970, or *Culiseta*: –0.61 (95% CI –1.83, 0.62), p-value = 0.3329, but for *Culex* mosquitoes, oviposition was significantly decreased by predator presence: –1.69 (95% CI –2.82,–0.56), p-value = 0.0033 (*Figure 4*).

## Discussion

In this study, laboratory and semi-field empirical data were obtained through a systematic literature review and used to conduct a meta-analysis that assessed consumptive and non-consumptive effects of predators on mosquito prey. Some results agree with previously observed trends, such as greater consumptive effects from larger predators (*Kumar et al., 2008*, *Peters, 1983*) and no oviposition response to predator cues among container-breeding *Aedes* mosquitoes (*Vonesh and Blaustein, 2010*). However, this meta-analysis revealed additional trends. Mosquito larval instar had an important role in moderating consumptive effects of predators, likely because of its direct correlation to prey size. Furthermore, a small, but significant, decrease in mosquito survival due to non-consumptive effects of predators was observed, suggesting that mosquitoes can be 'scared to death' by predators (*Preisser et al., 2005*). Both male and female body sizes were also reduced among mosquitoes that had been exposed to predators, and predator avoidance during oviposition was observed among female *Culex* mosquitoes. Effects of predators on different vector traits, particularly survival, body size, and oviposition behavior, have the potential to influence infectious disease dynamics.

### Consumptive effects of predators on survival

Several larger predators reduced mosquito survival, including freshwater fish (Cyprinidae and Poeciliidae), salamander larvae (Ambystomatidae), dragonfly larvae (Aeshnidae), and backswimmers (Notonectidae) (*Figure 3a*). This finding is consistent with a previous analysis which showed a positive linear relationship between predator body mass and ingestion rate across taxa (*Peters, 1983*). In addition, more effect size heterogeneity in the consumptive survival data was explained by an interaction between predator family and larval instar than was explained by an interaction between predator family and vector genus (*Table 3*). This result suggests that the relative sizes of predator and prey groups could play a more important role in determining consumptive mosquito survival than variations in predator responses to different behaviors of prey genera, which are likely to be shaped by the degree of shared evolutionary history between trophic levels (*Buchanan et al., 2017*). Larval instar is an indicator of mosquito size, and previous modeling work has provided evidence of prey size selection by predators to maximize energetic gain (*Mittelbach, 1981*). While smaller cyclopoid copepods are more effective against early instar mosquito larvae (*Dieng et al., 2002*), larger predators including

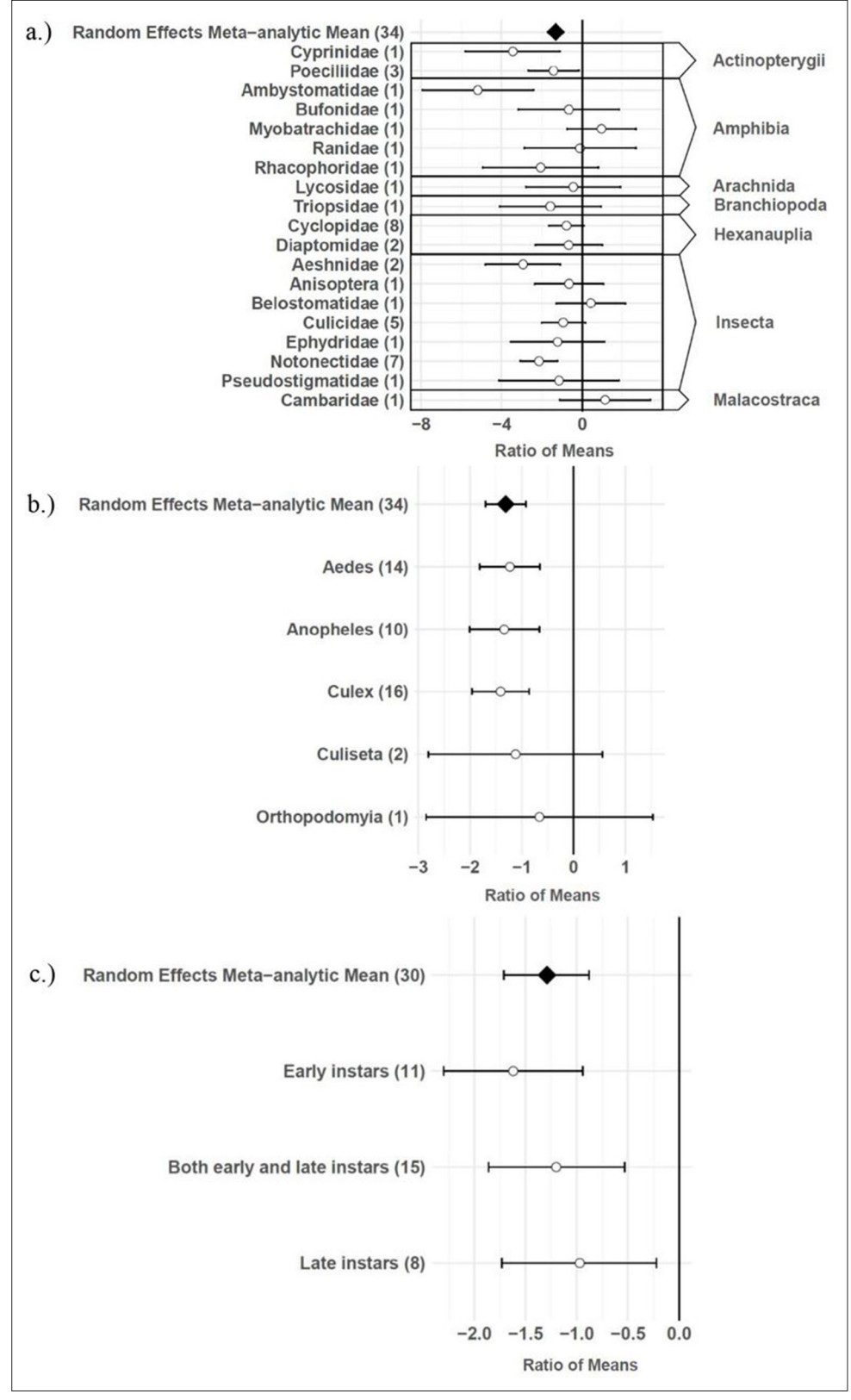

**Figure 3.** Effect sizes and 95 % confidence intervals for consumptive effects of predators, for different categories of moderators (with number of studies in parentheses).
(**a**) predator family with predator class in the right-hand column, (**b**) vector genus, and (**c**) larval instar.

**Table 2.** Candidate multilevel mixed effects models of consumptive effects from predators on mosquito survival, fitted to dataset of effect sizes (n = 187 from 34 studies), and ranked by corrected Akaike's information criterion (AICc).

| Moderator(s) | Test of moderators (degrees of freedom, p-value) | AICc | ΔAICc |
|---|---|---|---|
| Predator family x vector genus | 28, < 0.0001 | 500.5 | 0 |
| Predator family | 19, < 0.0001 | 507.0 | 6.5 |
| Predator family + vector genus | 23, < 0.0001 | 508.1 | 7.6 |
| Vector genus | 5, < 0.0001 | 573.0 | 72.5 |
| None | ---- | 576.5 | 76.0 |

**Table 3.** Candidate multilevel mixed effects models of consumptive effects from predators, fitted to dataset of effect sizes where larval instar is not missing (n = 163 from 30 studies), and ranked by corrected Akaike's information criterion (AICc).

| Moderator(s) | Test of moderators (degrees of freedom, p-value) | AICc | ΔAICc |
|---|---|---|---|
| Predator family x larval instar | 25, < 0.0001 | 429.2 | 0 |
| Predator family + larval instar | 19, < 0.0001 | 443.5 | 14.3 |
| Predator family x vector genus | 25, < 0.0001 | 455.0 | 25.8 |
| Predator family | 17, < 0.0001 | 456.8 | 27.6 |
| Predator family + vector genus | 21, < 0.0001 | 458.4 | 29.2 |
| Larval instar | 3, < 0.0001 | 503.1 | 73.9 |
| Vector genus | 5, < 0.0001 | 504.7 | 75.5 |
| None | ---- | 508.5 | 79.3 |

tadpoles, giant water bugs, dragonfly larvae, fish, and backswimmers are more effective against late instar larvae (*Kweka et al., 2011*).

## Non-consumptive effects of predators on survival

Exposure to predation cues significantly lowered mosquito survival, and this non-consumptive effect has also been observed in dragonfly larvae prey (*Leucorrhinia intacta*) that were exposed to caged predators (*McCauley et al., 2011*). The reduction in mosquito survival from non-consumptive effects of predators was significantly smaller than the reduction that was observed from consumptive effects. This is partially due to the practical constraints of most experimental designs, which cause consumptive and non-consumptive effects of predators on survival to be grouped together and reported as consumptive effects. The greater impact of combined consumptive and non-consumptive effects, in comparison to only non-consumptive effects, has previously been observed in pea aphids (*Acyrthosiphon pisum*) (*Nelson et al., 2004*).

## Non-consumptive effects of predators on body size

While predators did not significantly impact mosquito development time through non-consumptive effects in either sex, mosquito body size was decreased by the non-consumptive effects of predators in both sexes. Smaller body size is associated with lower reproductive success in mosquitoes because smaller females lay fewer eggs (*Blackmore and Lord, 2000*; *Lyimo and Takken, 1993*; *Oliver and Howard, 2011*; *Styer et al., 2007*; *Tsunoda et al., 2010*), and smaller males produce less sperm (*Hatala et al., 2018*; *Ponlawat and Harrington, 2007*). These effects suggest that predation could non-consumptively reduce mosquito population growth. The smaller size of mosquitoes exposed to predators could also limit disease transmission. Vector lifespan contributes disproportionately to disease transmission because older vectors are more likely to have been exposed to pathogens, more

likely to already be infectious after having survived the extrinsic incubation period, and more likely to survive long enough to bite subsequent hosts (*Cator et al., 2020*). It is well-established that smaller mosquito body size is associated with shorter mosquito lifespan (*Araújo et al., 2012*; *Hawley, 1985*, *Reisen et al., 1984*; *Reiskind and Lounibos, 2009*; *Xue et al., 2010*). Therefore, non-consumptive effects of predators may limit the transmission of mosquito-borne diseases.

## Non-consumptive effects of predators on oviposition behavior

Predator presence also non-consumptively reduced oviposition behavior in adult female mosquitoes. Meta-regression results showed that *Culex* females significantly avoid oviposition sites that contain predators or predator cues, but *Aedes* and *Culiseta* females do not avoid these sites, despite a slight non-significant trend toward predator avoidance in *Culiseta* (*Figure 4*). Both *Culex* and *Culiseta* mosquitoes have an 'all-or-none' oviposition strategy (*Johnson and Fonseca, 2014*), in which they lay hundreds of rapidly hatching eggs in rafts on the water's surface (*Day, 2016*). Such an oviposition strategy is conducive to evolving predator avoidance behaviors, and a previous meta-analysis showed significant predator avoidance in both *Culex* and *Culiseta* during oviposition (*Vonesh and Blaustein, 2010*). Conversely, it is likely that an oviposition response to predation is not particularly advantageous for *Aedes* because the delayed hatching of their eggs (*Day, 2016*) can prevent the level of predation risk at the time of oviposition from matching the level of predation risk present in the eventual larval environment (*Vonesh and Blaustein, 2010*). The predator avoidance response in *Aedes* species that lay their eggs above the water's edge in containers has previously been described as 'non-existent' (*Vonesh and Blaustein, 2010*). Both *Aedes* species included in this study's oviposition data subset,

**Table 4.** Candidate multilevel mixed effects models of non-consumptive effects of predators on mosquito oviposition behavior, fitted to dataset of effect sizes (n = 36 from 12 studies), and ranked by corrected Akaike's information criterion (AICc).

| Moderator(s) | Test of moderators (degrees of freedom, p-value) | AICc | ΔAICc |
|---|---|---|---|
| Vector genus | 3, 0.0149 | 122.1 | 0 |
| None | ---- | 125.2 | 3.1 |
| Predator family | 12, 0.8855 | 167.9 | 45.8 |

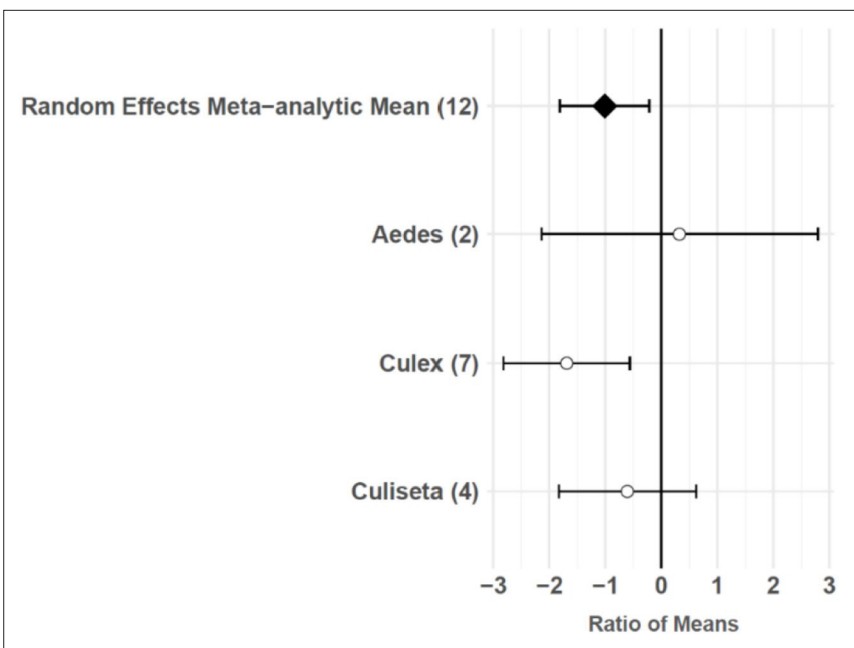

**Figure 4.** Oviposition effect sizes and 95 % confidence intervals for different categories of vector genus (with number of studies in parentheses).

*Ae. albopictus* and *Ae. aegypti*, meet the criterion of ovipositing above water in containers (*Juliano, 2009*). Predator avoidance during oviposition has previously been found to increase the mosquito population size at equilibrium (*Spencer et al., 2002*). However, this study's results and those of a previous meta-analysis (*Vonesh and Blaustein, 2010*) suggest that models of oviposition site selection, such as those using parameters from Notonectidae predators and *Culiseta* prey (*Kershenbaum et al., 2012*), are not generalizable to *Aedes* vectors.

## Implications for West Nile Virus disease dynamics

Predator avoidance during oviposition by *Culex* mosquitoes (*Figure 4*) may be of particular importance to West Nile virus (WNV) disease dynamics. Previous work has shown that *Cx. pipiens*, *Cx. restuans*, and *Cx. tarsalis* all avoid predator habitats (*Vonesh and Blaustein, 2010*), and that *Cx. pipiens* is the primary bridge vector of WNV responsible for spill-over transmission from avian reservoir hosts to humans (*Fonseca et al., 2004*; *Hamer et al., 2008a*, *Kramer et al., 2008*; *Andreadis, 2012*). *Cx. pipiens* mosquitoes can live in permanent aquatic environments, such as ground pools (*Amini et al., 2020*; *Barr, 1967*; *Dida et al., 2018*; *Sulesco et al., 2015*), ponds (*Lühken et al., 2015*), stream edges (*Amini et al., 2020*), and lake edges (*Vinogradova, 2000*) that are more common in rural areas, but *Cx. pipiens* are also found in urban and suburban residential areas, where they typically breed in artificial containers (*Sulesco et al., 2015*), including tires (*Lühken et al., 2015*; *Nikookar et al., 2017*; *Verna, 2015*), rainwater tanks (*Townroe and Callaghan, 2014*), and catch basins (*Gardner et al., 2012*). Small artificial containers, such as discarded tires, are generally unlikely to harbor larger predators, including freshwater fish (Cyprinidae and Poeciliidae), salamander larvae (Ambystomatidae), dragonfly larvae (Aeshnidae), and backswimmers (Notonectidae), because temporary aquatic environments cannot support the relatively long development times of these organisms. The mean dispersal distance of adult *Culex* mosquitoes is greater than one kilometer (*Ciota et al., 2012*; *Hamer et al., 2014*), and female *Cx. pipiens* have exhibited longer dispersal distances after developing in the presence of a fish predator (*Alcalay et al., 2018*). Therefore, predator avoidance during oviposition may cause *Cx. pipiens* populations to disperse from permanent aquatic environments in more rural areas to artificial container environments in urbanized areas, where the risk of human WNV infection is higher (*Brown et al., 2008*).

Predator cue levels may be altered by climate conditions, and these changes in cue levels can impact WNV transmission to humans. Drought has previously been associated with human WNV cases (*Johnson and Sukhdeo, 2013*; *Marcantonio et al., 2015*; *Roehr, 2012*; *Shaman et al., 2005*; *Epstein and Defilippo, 2001*; *Paull et al., 2017*), but the association has thus far lacked a clear underlying mechanism. Under drought conditions, the density of aquatic organisms increases and predation pressures can intensify due to compressed space and high encounter rates (*Amundrud et al., 2019*). A previous study of a stream ecosystem found that impacts of fish predation are more severe during the dry season (*Dudgeon, 1993*). In addition, reductions in water volume can facilitate consumption of mosquito larvae by crane fly larvae (Tipulidae), whereas mosquito consumption by tipulids was not observed at a higher water level (*Amundrud et al., 2019*). Laboratory and semi-field studies have shown that mosquitoes respond to a gradient of predator cues (*Roux et al., 2014*; *Silberbush and Blaustein, 2011*). The frequency of larval anti-predator behavior is correlated with the concentration of predator cues (*Roux et al., 2014*), and adult female mosquitoes prefer oviposition sites with lower predator densities (*Silberbush and Blaustein, 2011*). Therefore, as predator cue levels increase due to drought, permanent aquatic habitats are likely to transition from suitable oviposition sites for one generation of female mosquitoes, to unsuitable oviposition sites for the next generation.

When suitable oviposition sites are absent, females retain their eggs until sites become available (*Bentley and Day, 1989*). *Cx. pipiens* females can retain their eggs for up to five weeks, allowing them enough time to find container sites with low predation risk, often located in residential areas (*Johnson and Fonseca, 2014*). The movement of gravid female *Cx. pipiens* to residential areas increases the risk of WNV spill-over to humans because these vectors are likely to have already blood-fed at least once (*Clements, 1992*), suggesting that they have a higher risk of WNV infection, relative to non-gravid mosquitoes. This is consistent with studies that have reported associations between drought and WNV-infected mosquitoes in urban and residential areas (*Johnson and Sukhdeo, 2013*; *Paull et al., 2017*). In addition, vertical transmission of WNV from gravid females to their progeny may occur during oviposition (*Rosen, 1988*), when the virus is transmitted by an accessory gland fluid that attaches eggs

to one another (*Nelms et al., 2013*). Because the rate of vertical transmission in *Cx. pipiens* increases with the number of days following WNV infection (*Anderson et al., 2008*), extended searches for oviposition sites due to drought could increase the frequency of vertical transmission. However, the impact of vertical transmission on WNV epidemics is thought to be minimal because when transmission to an egg raft did occur, only 4.7% of the progeny were found to be infected as adults (*Anderson et al., 2008*), and only about half of those infected adults are estimated to be female. In summary, the movement of *Cx. pipiens* females toward more residential areas, combined with potential limited WNV amplification from increased vertical transmission, suggests that the vector trait of predator avoidance during oviposition can serve as a plausible explanation for associations between drought and human WNV cases.

Another theory for the association between drought and human WNV cases is based on the hypothesis that increased contact between mosquito vectors and passerine reservoir hosts occurs during drought conditions (*Paull et al., 2017*; *Shaman et al., 2005*). The proposed aggregation of bird and mosquito populations during drought was originally thought to occur in humid, densely vegetated hammocks – a type of habitat that is specific to southern Florida (*Shaman et al., 2005*), but WNV incidence is more consistently clustered in other regions of the US, particularly the Northern Great Plains (*CENTERS FOR DISEASE CONTROL AND PREVENTION, 2021*; *Sugumaran et al., 2009*). Northern cardinals (*Cardinalis cardinalis*), American robins (*Turdus migratorius*), and house sparrows (*Passer domesticus*) were among the bird species that most frequently tested seropositive for WNV antibodies in 2005 and 2006 in Chicago, where high numbers of human cases were reported (*Hamer et al., 2008b*), and these passerine species are more abundant in residential areas, regardless of precipitation patterns (*Anderson, 2006b*; *Beddall, 1963*; *Lepczyk et al., 2008*). Apart from drought, landowners' participation in supplemental bird feeding, providing bird houses, gardening, and maintaining vegetation can strongly influence passerine abundance in residential areas (*Lepczyk et al., 2004*). Furthermore, as terrestrial foragers that can obtain hydration from their diet of insects, fruits, and other plant material (*Anderson, 2006a*; *Brzek et al., 2009*; *Malmborg and Willson, 1988*; *Renne et al., 2000*), passerine reservoir hosts of WNV are less likely to move in response to drought than the mosquito vectors of WNV, which have obligate aquatic life stages.

While hatch-year birds are more vulnerable to mosquito biting, and thus contribute to the amplification of WNV (*Hamer et al., 2008b*), it is illogical to expect an increased abundance of hatch-year birds during drought conditions. However, some have argued that in cases where drought decreases the abundance of juvenile birds, the ratio of mosquitoes to birds increases, and this could lead to higher WNV prevalence in the mosquito population (*Paull et al., 2017*). Although reductions in both hatching success (*George et al., 1992*) and survival of recently fledged birds (*Yackel Adams et al., 2006*) have been observed during drought conditions, the impact of drought on avian abundance varies widely by species (*Verner and Purcell, 1999*). In particular, synanthropic species, such as those likely to harbor WNV, are less negatively affected by drought (*Albright et al., 2009*). Additionally, the droughts that impact avian abundance often occur over much longer periods of time than the seasonal droughts that predict WNV transmission to humans. For example, avian abundance has been modeled based on precipitation metrics spanning 32 weeks, and house wren (*Troglodytes aedon*) abundance has been predicted by precipitation averages spanning four years (*Verner and Purcell, 1999*). Finally, birds with higher levels of stress hormones are more likely to be fed on by mosquitoes, and certain factors associated with residential areas, such as road noise, light pollution, and pesticide exposure, can cause avian stress (*Gervasi et al., 2016*). Therefore, elevated avian stress hormones in these habitats may contribute to WNV prevalence in the mosquito population, independent of drought conditions.

## Implications for mosquito-borne disease modeling

Although the aquatic phase of the mosquito life cycle is often overlooked in mathematical models of mosquito-borne pathogen transmission (*Reiner et al., 2013*), vector survival at immature stages plays an important role in determining mosquito population abundance, which is an essential factor for predicting disease transmission (*Beck-Johnson et al., 2013*). The results of this study show that mosquito survival decreases among the *Aedes*, *Anopheles*, and *Culex* genera due to consumptive effects of predators (*Figure 3b*), and that there is also a reduction in mosquito survival due to non-consumptive effects. Other studies have demonstrated that aquatic predators dramatically impact

mosquito survival and abundance. For example, a biocontrol intervention relying on the application of copepod predators eliminated *Aedes albopictus* from three communes in Nam Dinh, Vietnam, where dengue transmission was previously detected, and reduced vector abundance by 86–98% in three other communes (*Kay et al., 2002*). Conversely, the annual abundance of *Culex* and *Anopheles* mosquitoes was observed to increase 15-fold in semi-permanent wetlands in the year following a drought, likely because the drought eliminated aquatic predators from wetlands that dried completely, and mosquitoes were able to re-colonize newly formed aquatic habitats more quickly than their most effective predators (*Chase and Knight, 2003*).

While relationships between temperature and different vector traits, such as fecundity and lifespan, have been incorporated into models of temperature effects on mosquito population density (*El Moustaid and Johnson, 2019*), models of predator effects on vector borne disease transmission have focused primarily on the impacts of predation on vector survival. Previous models have shown that predators of vector species can decrease or eliminate pathogen infection in host populations as vector fecundity increases (*Moore et al., 2010*). The findings of this meta-analysis suggest that predators also decrease vector fecundity through non-consumptive effects on vector body size. In addition, the entomological inoculation rate (EIR) is likely to be reduced by effects of predators on mosquito fecundity and lifespan, as well as effects of predators on mosquito survival. The EIR has been defined as the product of three variables: ($m$) the number of mosquitoes per host, ($a$) the daily rate of mosquito biting, and ($s$) the proportion of mosquitoes that are infectious (*Beck-Johnson et al., 2013*). Based on this study's findings, predators are likely to decrease the number of mosquitoes per host by reducing mosquito survival through both consumptive and non-consumptive effects, and by reducing mosquito fecundity through non-consumptive effects on body size. In addition, predators are likely to decrease the proportion of mosquitoes that are infectious by shortening the vector lifespan through non-consumptive effects on body size. The relationship between mosquito body size and biting rate is unclear, with some studies showing higher biting rates among larger mosquitoes (*Araújo et al., 2012*; *Gunathilaka et al., 2019*), and others reporting higher biting rates among smaller mosquitoes (*Farjana and Tuno, 2013*; *Leisnham et al., 2008*). The links between factors that influence the EIR and observed effects of predators on mosquito prey demonstrate the necessity of including both consumptive and non-consumptive effects of predators in models of mosquito-borne disease.

## Conclusion

This meta-analysis on mosquito predation demonstrates that predators not only play an important role in directly reducing mosquito populations, but also have non-consumptive effects on surviving mosquitoes that may ultimately reduce further population growth and decrease disease transmission. While families of larger sized predators were effective in reducing mosquito survival, other factors, such as impacts on native species, as well as the economic cost of mass-rearing and field applications (*Kumar and Hwang, 2006*; *Pyke, 2008*), should be carefully considered before selecting a predator as a suitable biocontrol agent. Predictive disease models are likely to be more reliable when the non-consumptive effects of predation are incorporated. Although exposure of mosquito larvae to predators is commonplace in outdoor field settings, it remains rare in most laboratory-based assessments of vector traits. Therefore, mosquitoes observed in nature are likely to have smaller body sizes than those observed under optimal laboratory conditions. It is important for disease modelers to recognize these impacts of predation on vector traits as they can reduce mosquito population growth and limit disease transmission due to shorter vector lifespans. Within the WNV disease system, consideration of the oviposition behavioral response to predation cues by *Culex* vectors can improve current understanding of the association between drought and human cases. This study provides general estimates of the effects of predators on selected mosquito traits for use in predictive disease models.

## Future directions

Modeling efforts that aim to optimize the application of biocontrol predators should also consider incorporating predator effects on vector survival, fecundity, and lifespan. These additions to predictive models of various biocontrol interventions are likely to help public health officials choose the most cost-effective strategies for limiting disease transmission. In the 60-study database that was compiled, only one study was designed to directly measure the effect of larval-stage predation on vector competence (*Roux et al., 2015*). Therefore, future efforts to assess the impact of predators

on mosquito-borne disease transmission should prioritize experimental studies in which infected mosquito larvae are observed throughout an initial period of aquatic exposure to predators, followed by a period of blood-feeding in the adult stage.

Two studies from the compiled database examined the compatibility of predators with *Bacillus thuringiensis* var. *israelensis* (*Bti*), a commonly used bacterial biocontrol agent (*Chansang et al., 2004*; *Op de Beeck et al., 2016*). Previous studies have supported the simultaneous application of cyclopoid copepod predators and *Bti* (*Marten et al., 1993*; *Tietze et al., 1994*), but additional analyses are needed on the use of *Bti* with other families of mosquito predators. Populations of other insect pests, such as the southern green stink bug (*Nezara viridula*), are known to be regulated by both predators and parasites (*Ehler, 2002*). The literature search conducted for this meta-analysis returned studies on water mite parasites (*Rajendran and Prasad, 1994*) and nematode parasitoids (*de Valdez, 2006*) of mosquitoes, and ascogregarine parasites have previously been evaluated as biocontrol agents against *Aedes* mosquitoes (*Tseng, 2007*). A more thorough review of the impacts of parasites and parasitoids on vector traits, such as survival, fecundity, and lifespan, is needed before incorporating these potential biocontrol agents into integrated vector control plans.

Three studies in the 60-study database included experiments where two mosquito prey species were made available to the predator species (*Grill and Juliano, 1996*; *Griswold and Lounibos, 2005*, *Micieli et al., 2002*). In these cases, the effect size measurement for each mosquito species could be influenced by interspecific competition, or a preference of the predator species for a certain prey species. Hetero-specific prey observations were excluded from this meta-analysis, but future analyses centered on the concepts of interspecific competition or predator preferences might further evaluate these data. In addition, this meta-analysis investigated consumptive and non-consumptive effects of predators separately. More research is needed to determine how models should combine these different types of predator effects to accurately reflect predation interactions as they occur in natural environments.

## Acknowledgements

We are grateful to Dr. Lauren Cator for facilitating the early stages of this project and helping to edit this manuscript. We also thank Dr. Peter Hudson for his helpful advice in early discussions about the project's aims. In addition, we thank all members of the VectorBiTE RCN (https://vectorbite.org/) for taking the initiative to forge productive collaborations between empiricists and modelers in vector ecology. This work was funded by NIH grant 1R01AI122284-01 and BBSRC grant BB/N013573/1 as part of the joint (NIH-NSF-USDA-BBSRC) Ecology and Evolution of Infectious Diseases program. It was also funded by a President's PhD Scholarship from Imperial College London awarded to Marie C Russell.

## Additional information

### Funding

| Funder | Grant reference number | Author |
| --- | --- | --- |
| National Institutes of Health | 1R01AI122284-01 | Zachary Gajewski<br>Fadoua El Moustaid |
| Biotechnology and Biological Sciences Research Council | BB/N013573/1 | Zachary Gajewski<br>Fadoua El Moustaid |
| Imperial College London | President's PhD Scholarship | Marie C Russell |

The funders had no role in study design, data collection and interpretation, or the decision to submit the work for publication.

### Author contributions

Marie C Russell, Conceptualization, Data curation, Formal analysis, Investigation, Methodology, Project administration, Software, Validation, Visualization, Writing – original draft, Writing – review

and editing; Catherine M Herzog, Andrew C McCall, Conceptualization, Data curation, Formal analysis, Investigation, Methodology, Project administration, Software, Supervision, Validation, Visualization, Writing – review and editing; Zachary Gajewski, Michelle V Evans, Conceptualization, Data curation, Formal analysis, Investigation, Methodology, Software, Validation, Writing – review and editing; Chloe Ramsay, Conceptualization, Data curation, Formal analysis, Investigation, Methodology, Validation, Writing – review and editing; Fadoua El Moustaid, Conceptualization, Data curation, Writing – review and editing; Trishna Desai, Conceptualization, Data curation, Visualization, Writing – review and editing; Nicole L Gottdenker, Sara L Hermann, Conceptualization, Data curation, Supervision, Writing – review and editing; Alison G Power, Conceptualization, Supervision, Writing – review and editing

**Author ORCIDs**
Marie C Russell ⓘ http://orcid.org/0000-0001-6907-6159
Catherine M Herzog ⓘ http://orcid.org/0000-0003-3021-888X

**Decision letter and Author response**
Decision letter https://doi.org/10.7554/eLife.71503.sa1
Author response https://doi.org/10.7554/eLife.71503.sa2

## Additional files

### Supplementary files
• Supplementary file 1. Table 1: Publications included in the database and their types of vector trait data.
• Transparent reporting form

### Data availability
The database can be accessed here: https://doi.org/10.5061/dryad.4qrfj6q9x. The R code file, showing all analyses, can be accessed here: https://doi.org/10.5281/zenodo.5790092.

The following dataset was generated:

| Author(s) | Year | Dataset title | Dataset URL | Database and Identifier |
|---|---|---|---|---|
| Russell MC, Herzog CM, Gajewski Z, Ramsay C, El Moustaid F, Evans M, Desai T, Gottdenker N, Hermann S, Power A, McCall A | 2021 | Both consumptive and non-consumptive effects of predators impact mosquito populations and have implications for disease transmission | https://doi.org/10.5061/dryad.4qrfj6q9x | Dryad Digital Repository, 10.5061/dryad.4qrfj6q9x |

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
