## [Editor Report]

This careful meta-analysis evaluates consumptive and non-consumptive effects of aquatic predators across multiple mosquito species, drawing from laboratory and semi-field studies. The authors find an important role for larval size in moderating consumption, significant non-consumptive impacts of predators on survival and body size, and variable effects of predators on oviposition behavior. These results therefore highlight multiple mechanisms by which aquatic predators might affect disease transmission.

---

## [Decision Letter]

**Decision letter after peer review:**

Thank you for submitting your article "Both consumptive and non-consumptive effects of predators impact mosquito populations and have implications for disease transmission" for consideration by *eLife*. Your article has been reviewed by two peer reviewers, and the evaluation has been overseen by a Reviewing Editor and Christian Rutz as the Senior Editor. The following individual involved in the review of your submission has agreed to reveal their identity: Amanda Meadows (Reviewer #2).

The reviewers have discussed their reviews with one another, and the Reviewing Editor has drafted this decision letter to help you prepare a revised submission.

Essential revision: Both reviewers agreed in consultation that the discussion of WNV seems to overreach the results of the study. Please revise this section.

Optional: Please also address the clarification requests listed in the reviewers' reports below.

We thank the authors for the clear, well-documented code and description of the data file.

*Reviewer #1 (Recommendations for the authors):*

Line 82: The authors refer to experimental observations of predator effects on mosquito development and size as 'inconsistent'. While they refer to the sex-specific effects, what are their remarks on 'contextual or interactive effects ' driven by conspecific/heterospecific larval densities, for instance?

Line 170: It is unclear what "development time predator" means here.

Finally, the consumptive and non-consumptive effects of predation are known to interact with factors such as larval density or inter/intraspecific competition. There is experimental evidence of larval mortality resulting from consumptive effects leading to an increase in the population size of mosquitoes. I do not see this study analyzing the significance of such complex effects involving predation risk. The selection criteria employed by the authors to build the vector trait data subsets might have favored the bias towards studies investigating lone effects of predation threat. How can the differential influences of consumptive and non-consumptive effects of predators on mosquito traits be put to better use in disease models? If possible, the authors could consider discussing this aspect briefly in the manuscript.

*Reviewer #2 (Recommendations for the authors):*

Most of my suggestions refer to the "Implications for West Nile virus disease dynamics" section in the discussion.

Lines 539-562 This largely seems to be describing the effect of drought on WNV transmission dynamics, which seems to have very little to do with predator NCEs or consumptive effects. This is valuable space that could be used discussing the implications of traits shown to be impacted by predators in this study (such as body size). The authors do mention drought is known to move mosquitoes into more urban areas in search of breeding sites, but one important known driver is the consolidation of birds in mosquito breeding sites searching for water and potentially a high vector to reservoir ratio through a die off of juvenile birds (see https://www.ncbi.nlm.nih.gov/pmc/articles/PMC5310598/, for example). I think if the authors decide to keep this section, these alternatives should be discussed as well.

---

## [Author Response]

Reviewer #1 (Recommendations for the authors):Line 82: The authors refer to experimental observations of predator effects on mosquito development and size as 'inconsistent'. While they refer to the sex-specific effects, what are their remarks on 'contextual or interactive effects ' driven by conspecific/heterospecific larval densities, for instance?

Because only three studies in our database included hetero-specific mosquito prey, examining these interactions was deemed to be outside the scope of our meta-analysis. We now include three sentences in the “Future directions” section (Lines 648-654) that address the need for more research on this topic.

Line 170: It is unclear what "development time predator" means here.

We acknowledge that this sentence was not structured well, and we have revised it to improve its level of clarity (Line 170-171).

Finally, the consumptive and non-consumptive effects of predation are known to interact with factors such as larval density or inter/intraspecific competition. There is experimental evidence of larval mortality resulting from consumptive effects leading to an increase in the population size of mosquitoes. I do not see this study analyzing the significance of such complex effects involving predation risk. The selection criteria employed by the authors to build the vector trait data subsets might have favored the bias towards studies investigating lone effects of predation threat. How can the differential influences of consumptive and non-consumptive effects of predators on mosquito traits be put to better use in disease models? If possible, the authors could consider discussing this aspect briefly in the manuscript.

We are aware of certain instances where the consumptive and non-consumptive effects of predators seem to have opposing impacts on mosquito populations. For example, in a low food resource environment, consumption of some mosquito larvae could release the surviving larvae from the constraints of intraspecific competition, increasing the per capita food intake and allowing survivors to grow larger in size; these larger mosquitoes would likely have greater reproductive potential than those that developed in the absence of predation. Some experiments in our database were carefully designed to only examine non-consumptive effects; these experiments often expose mosquito larvae to a caged predator, or to predator cues that do not include a live predator. Other experiments in our database that measure non-consumptive effects do allow the predator to have direct access to mosquito prey. To examine how different types of predator cues might alter non-consumptive effects on mosquito body size and development time, we included two variables, number of predators (Num_pred) and type of cue (Cue_type), in our database. In cases where the predator(s) did not have direct access to prey, the Num_pred variable is listed as “CUE”. The Cue_type variable differentiates between cues that are only from the predator(s), and cues that include both the predator(s) and dying mosquito prey. Unfortunately, none of the four size or development data subsets met all the criteria that we required to test for moderators (Lines 261-264). Thus, investigating how different predator cue types could moderate the non-consumptive effects of predators on mosquito size or development time is beyond the capabilities of this meta-analysis.

Many past studies in the literature analyze consumptive and non-consumptive effects of predators separately. Our selection criteria did not create this separation, but our effect size calculations did limit our database to observations from manipulative experiments. An observational field study would be the most accurate study design for assessing how consumptive and non-consumptive effects combine in natural environments, but data from both predator and control treatments, as generated in manipulative experiments, are needed to determine effect sizes. To acknowledge the need for further evaluation of how consumptive and non-consumptive effects combine in natural settings, we have added two sentences to the “Future directions” section (Lines 654-657).

Reviewer #2 (Recommendations for the authors):Most of my suggestions refer to the "Implications for West Nile virus disease dynamics" section in the discussion.Lines 539-562 This largely seems to be describing the effect of drought on WNV transmission dynamics, which seems to have very little to do with predator NCEs or consumptive effects. This is valuable space that could be used discussing the implications of traits shown to be impacted by predators in this study (such as body size). The authors do mention drought is known to move mosquitoes into more urban areas in search of breeding sites, but one important known driver is the consolidation of birds in mosquito breeding sites searching for water and potentially a high vector to reservoir ratio through a die off of juvenile birds (see https://www.ncbi.nlm.nih.gov/pmc/articles/PMC5310598/, for example). I think if the authors decide to keep this section, these alternatives should be discussed as well.

Insect oviposition behavior has previously been discussed as a trait that can be non-consumptively affected by predators, and limited mosquito oviposition sites during drought conditions has previously been identified as a potential cause of the association between drought and human WNV cases. Because our goal of connecting the meta-analysis findings to infectious disease dynamics is stated in the Introduction section, we maintain that our discussion of how drought could intensify the non-consumptive effects of predators on mosquito oviposition behavior, in the context of the WNV disease system, is deserving of space in this manuscript. However, we have edited the original WNV paragraphs (Lines 451-516) to eliminate approximately 70 words that unnecessarily contributed to the length of this section.

In addition, we have taken several steps to ensure that our discussion points on this topic are not overstated. Instead of claiming that predator avoidance during oviposition “is the behavioral mechanism underlying associations between drought and human WNV cases”, we now suggest that this trait “can serve as a plausible explanation” for these previously observed associations (Lines 514-516). We also revised the part of the Conclusion related to this topic from “can greatly improve current understanding”, to “can improve current understanding” (Line 618). Furthermore, we have removed mention of “paradoxical” and “counter-intuitive” as descriptors of the association between drought and WNV cases (previously included around Lines 479-480). Instead, we now include discussion of alternate theories, as well as discussion of several weaknesses that are apparent within these theories (Lines 518-557).